# Learning Mutation-Aware Visual Context for Antibody-Antigen Affinity Maturation Prediction

## Abstract

Modeling the impact of amino acid mutations on antibody–antigen binding affinity is critical for therapeutic antibody design. Existing structure-based deep learning approaches can capture structural details of binding interfaces, but they often fail to account for subtle physicochemical perturbations introduced by mutations, limiting their ability to explain affinity shifts. To address this challenge, we present ImageAM, a mutation-aware vision transformer framework that learns from unlabeled protein-protein interaction ground-truth data. ImageAM projects multiple structural and physicochemical interface features into two-dimensional (2D) images and employs a multi-channel masked reconstruction pretraining task, enabling the model to learn mutation-induced patterns across heterogeneous contexts. This pretraining strategy equips the encoder with strong generalization ability, which is further refined through fine-tuning for antibody affinity maturation prediction. Extensive experiments on benchmark datasets demonstrate that ImageAM consistently surpasses state-of-the-art methods across multiple metrics, while exhibiting superior robustness and out-of-distribution generalization in predicting binding affinity change between mutant and wild-type complexes. Code is available at `https://anonymous.4open.science/r/ImageAM-ICLR`.

## 1 Introduction

Biological functions are carried out through molecular interactions and chemical reactions, among which protein–protein interactions represent one of the most fundamental molecular events in living organisms. Many essential biological processes are mediated by protein interactions Jones & Thornton (1996); Mendelsohn & Brent (1999). A particularly important example is the interaction between antibodies and antigens. Antibodies, as immune system proteins, play a central role in human immunity by binding to target antigens and triggering immune responses Lu et al. (2018). The binding affinity of antibody–antigen interactions is a key indicator of the strength and effectiveness of this immune recognition Lu et al. (2020b). Mutations of amino acids at the binding interface frequently alter binding affinity, which may either enhance, weaken, or even disrupt the antibody–antigen interaction Gram et al. (1992). The process of affinity maturation refers to the gradual improvement of antibody binding efficacy to antigens through somatic hypermutation in vivo Victora & Nussenzweig (2022). In computational studies, affinity maturation is commonly defined as the change in binding free energy ($\Delta\Delta G = \Delta G_{mt} - \Delta G_{wt}$, where $mt$ and $wt$ denote the mutant and wild-type, respectively).

Traditional experimental approaches, such as constructing mutant libraries and screening with display technologies Li et al. (2014), are capable of characterizing antibody affinity maturation. However, the vast diversity of antibody–antigen interface sequences, combined with the combinatorial explosion of possible mutation types and positions, makes exhaustive exploration of the mutational landscape infeasible. Experimental strategies alone often struggle to balance comprehensiveness and efficiency Alves (2019). Moreover, these assays are resource-intensive and time-consuming, making them difficult to scale for rapid iteration and large-scale applications. Consequently, there is a pressing need for computational methods that can achieve both accuracy and efficiency, thereby accelerating research on antibody affinity maturation.

Recent advances in machine learning, particularly deep learning, have shown promising results in this task. Early sequence-based models treated antibody and antigen amino acid sequences as inputs, employing architectures such as convolutional neural networks (CNNs) LeCun et al. (1995), recurrent neural networks (RNNs) Quang & Xie (2016), and Transformers Vaswani (2017). More recently, structure-based methods have demonstrated greater potential by representing antibody–antigen binding interfaces as three-dimensional graphs and applying graph neural networks (GNNs) Scarselli et al. (2008). While structure-based methods have shown promise in modeling antibody–antigen interfaces under different mutations, they often struggle to capture subtle physicochemical perturbations such as changes in hydropathy, hydrogen bonding, or electrostatics that critically influence binding affinity. As illustrated in Figure 1, even minor point mutations can leave the backbone structure largely intact (left) while inducing significant shifts in charge and hydropathy distributions at the interface (right). Such effects are difficult to detect with purely geometric or graph-based models Yamashita et al. (2019); Jin & Wells (1994). In contrast, vision-based modeling provides a stronger perceptual capacity to represent fine-grained, spatially distributed changes across multiple physicochemical channels.

Motivated by this, we propose ImageAM, a mutation-aware unsupervised framework that learns robust visual representations of interface perturbations. Specifically, we describe antibody–antigen binding interfaces as surface molecular fingerprints that integrate both geometric and physicochemical properties, including curvature, charge, and hydropathy, etc. These features are projected within a defined neighborhood of the interaction center into multi-channel 2D images Stebliankin et al. (2023), enabling the use of high-capacity vision transformer (ViT) architectures in this domain. To effectively capture mutation-aware representations, we introduce a multi-channel masked image modeling pretraining task inspired by masked autoencoders (MAE) He et al. (2022). This design enables the model to reconstruct masked patches across different feature channels, leveraging cross-channel correlations

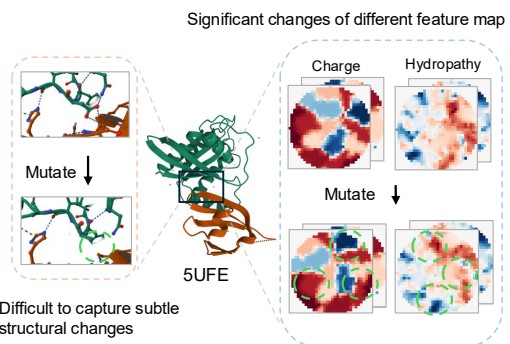

Figure 1: Impact of Subtle Mutations on Hydropathy and Charge at the Contact Interface of Protein 5UFE. The green dashed circles in the figure highlight the changes exhibited by the mutation.

and learning the impact of local perturbations on global interface context. The pretrained encoder is then partially frozen and fine-tuned on labeled data to predict changes in binding free energy ($\Delta\Delta G$) for antibody–antigen mutations.

In summary, our work makes the following contributions: (1) **A novel representation**. We introduce the first framework that encodes antibody–antigen binding interfaces as surface molecular fingerprint images, which naturally integrate structural and physicochemical features and open the door to applying powerful vision-based architectures in this domain. (2) **A pretrainable architecture**. We propose ImageAM, a vision transformer with MAE-style multi-channel masked image modeling, enabling effective pretraining under limited data and enhancing generalization from local mutations to global binding interface properties. (3) **Strong empirical performance**. Extensive experiments demonstrate that ImageAM consistently outperforms strong baselines across multiple benchmarks for $\Delta\Delta G$ prediction.

## 2 RELATED WORK

### 2.1 MUTATIONAL EFFECT PREDICTION

Traditional computational approaches for affinity prediction are primarily based on empirical energy functions. These methods estimate binding affinity by sampling conformations of protein–ligand complexes and applying energy functions derived from classical mechanics or statistical potentials Schymkowitz et al. (2005); DeBartolo et al. (2014); Steinbrecher et al. (2017).

With the rise of machine learning, increasingly more ML-based methods have emerged. Conventional machine learning approaches typically integrate geometric, physical, and evolutionary features of complexes to fit experimental data Steinbrecher et al. (2017); Zhang et al. (2022). However, these traditional computational strategies often struggle to balance efficiency and accuracy. In recent years, the rapid development of deep learning has led to significant progress in this field. Broadly, deep learning models can be categorized into sequence-based and structure-based approaches.

Sequence-based models leverage architectures such as CNNs LeCun et al. (1995), RNNs Quang & Xie (2016), and Transformers Vaswani (2017) to extract predictive features from antibody and antigen amino acid sequences. More recently, large-scale protein language models such as ESM Rives et al. (2021) and Saprot Su et al. (2023) have demonstrated strong performance, benefiting from pretraining on massive protein sequence corpora that enable them to capture rich contextual information.

In contrast, structure-based methods typically represent the three-dimensional geometry of the binding interface as graphs, allowing models to exploit more detailed and fine-grained structural information than sequence alone. For instance, Bind-ddG Cao et al. (2019) predicts binding changes by separately encoding residue-pair information at the antibody–antigen interface. RDE-Network Luo et al. (2023) extracts both single-residue and residue-pair representations and predicts mutational effects by modeling conformational flexibility at the interface. GearBind Cai et al. (2024) employs a multi-level geometric GNN to encode interface structures and achieve accurate predictions.

## 2.2 Pretraining on Proteins

To address the scarcity of labeled protein data, a growing body of work has focused on self-supervised and unsupervised pretraining strategies to extract transferable protein representations and enhance downstream task performance. These approaches can also be broadly divided into sequence-based and structure-based paradigms.

Sequence-based pretraining treats amino acids as tokens in a sequence and adapts paradigms such as Masked Language Modeling Rives et al. (2021); Su et al. (2023), Contrastive Learning Lu et al. (2020a), and Next-Token Prediction Alley et al. (2019). While sequence pretraining has achieved remarkable success, structure-aware pretraining is attracting increasing interest, as protein function is largely dictated by three-dimensional conformation.

Structure-based pretraining methods leverage tasks such as Masked Structure Modeling Wu et al. (2024), Structure Contrastive Learning Zhang et al. (2022), and Inter-residue Geometry Prediction Chen et al. (2023) to capture structural representations that generalize effectively across downstream applications.

## 3 Method

### 3.1 Problem Statement

Given the structures of an antibody-antigen mutant and its corresponding wild-type, our goal is to predict the affinity maturation of the mutant relative to the wild-type. This is quantified as the binding free energy change $\Delta\Delta G$. The antibody-antigen interface is represented by a set of multi-channel projection images that encode complementary physicochemical features, including shape complementarity, relative accessible surface area (RASA), hydrogen bonds, charge distribution, and hydropathy.

Formally, the interface structure is represented as a tensor input $\mathbf{X} \in \mathbb{R}^{C \times L \times L}$, where $C$ denotes the number of feature channels and $L$ is the width of the projection patch at the interface. The inputs include the wild-type tensor $\mathbf{X}^w$ and the mutant tensor $\mathbf{X}^m$. The task is to learn a function:

$$f : (\mathbf{X}^w, \mathbf{X}^m) \mapsto y,$$

where $y \in \mathbb{R}$ denotes the experimentally measured binding affinity change ($\Delta\Delta G$) associated with the mutation. Our objective is to design a pretraining and fine-tuning strategy that enables the model to extract discriminative and robust interface representations from $(\mathbf{X}^w, \mathbf{X}^m)$, ultimately improving the prediction accuracy of binding affinity changes.

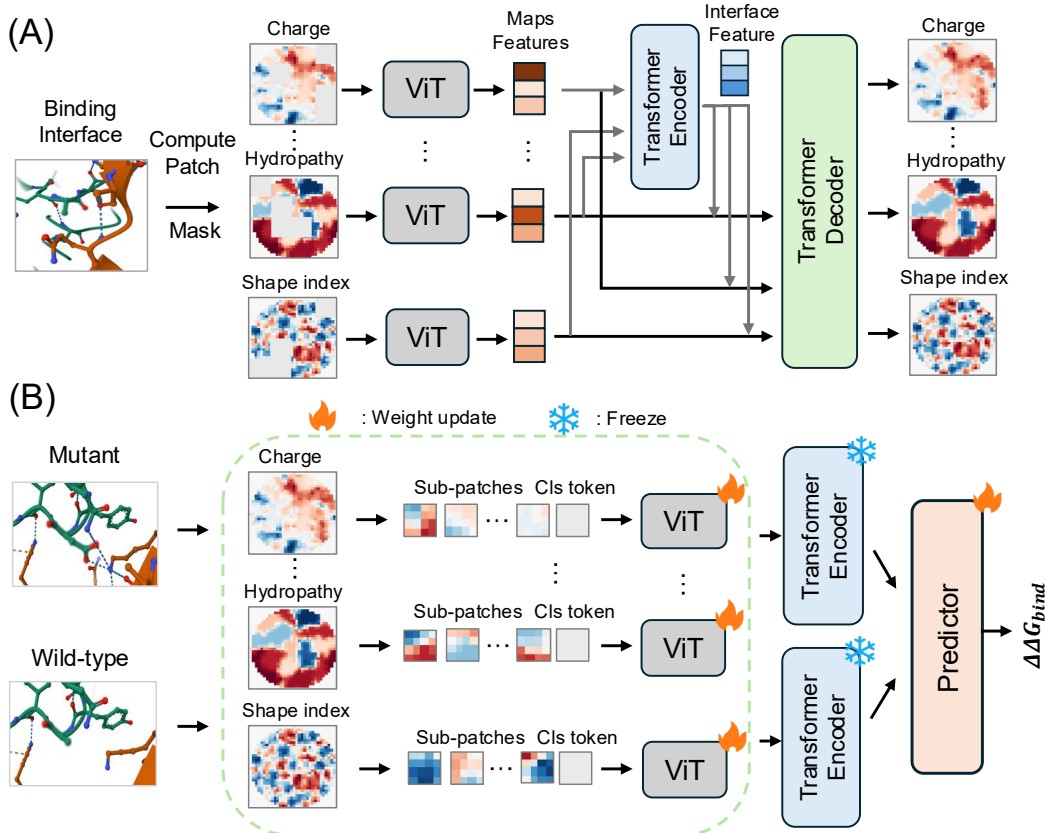

Figure 2: **(A) Multi-channel Masked Reconstruction Pretraining**. The masked multi-channel images are first encoded by a ViT to extract channel-wise features. A Transformer encoder then integrates features across channels to obtain the global representation. Finally, each channel is reconstructed based on its own features and the fused representation. **(B)** $\Delta\Delta G$ **Prediction**. We extract the encoder from the pretrained model, freeze part of its structure, and fine-tune the remaining components on an antibody–antigen affinity dataset to predict the $\Delta\Delta G$ values.

## 3.2 OVERVIEW

An overview of the proposed pretraining framework and affinity maturation prediction model is shown in Figure 2. In this section, we first introduce the **Data Preparation** procedure. Unlike other structure-based methods, we represent the antibody-antigen interface as multi-channel feature images. We then describe the proposed pretraining method: **Multi-Channel Vision Transformer Encoding** extracts representations from each channel, **Cross-Channel Feature Aggregation** fuses them into global features, and **Masked Token Reconstruction** reconstructs masked patches for self-supervised learning. Finally, in the $\Delta\Delta G$ **Prediction** stage, we show how the pretrained encoder is fine-tuned for predicting binding affinity maturation.

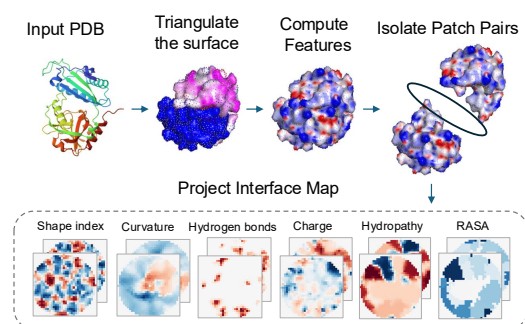

Figure 3: Data preparation process for binding interface representation.

### 3.3 DATA PREPARATION

Unlike conventional structure-based approaches that model antibody-antigen interfaces as graphs, our input modality is a set of multi-attribute projection images of the interface surface. Preprocessing of the structural data is required before model input.

As shown in Figure 3, to reduce computational cost, we first crop the input structures within a radius of 20 Å from the antibody-antigen interaction center. Atoms from antibody and antigen with interatomic distances less than 5 Å are defined as interaction sites. The cropped structures are triangulated into solvent-excluded surfaces at 1 Å resolution using MaSIF Gainza et al. (2020). We then compute vertex-level features including shape index, curvature, hydrogen-bond potential, charge, and hydropathy via the MaSIF data preparation module. In addition, relative accessible surface area (RASA) is obtained using DSSP v2.3 Kabsch & Sander (1983); Touw et al. (2015).

Finally, we extract a 20 Å radius patch centered at the interaction site. The structural and physicochemical features of the surface within the patch are projected into 2D images Stebliankin et al. (2023), where pixel intensity is proportional to the corresponding feature values.

### 3.4 MULTI-CHANNEL MASK RECONSTRUCTION PRETRAINING

To address the scarcity of labeled protein data, an increasing number of self-supervised or unsupervised models have been developed to extract transferable protein representations, thereby enhancing performance on specific downstream tasks. Inspired by Masked Autoencoders (MAE) He et al. (2022), we design a multi-channel masked reconstruction pretraining task to capture the distributional characteristics of protein–protein interaction interfaces. By partially masking and reconstructing different interface features, the model learns how local variations influence global representations, enabling it to capture the impact of subtle mutations on the overall interface. During decoding, each channel is reconstructed while considering fused information from all channels, which equips the model with the ability to account for cross-channel interactions. This design allows the model to better integrate diverse features when handling mutations in downstream tasks, thereby enhancing its representational power.

#### 3.4.1 MULTI-CHANNEL VISION TRANSFORMER ENCODING

Let the input interface image be $\mathbf{X} \in \mathbb{R}^{B \times C \times L \times L}$, where $B$ is the batch size, $C$ the number of feature channels, and $L \times L$ the spatial resolution of each channel.

For each channel $c$, we extract the subset $\mathbf{X}_c \in \mathbb{R}^{B \times C_c \times L \times L}$. Each channel image is divided into $N$ non-overlapping patches, flattened, and projected into token embeddings:

$$\mathbf{T}_c^0 = [t_1^c, t_2^c, \ldots, t_N^c, t_{\text{cls}}^c] \in \mathbb{R}^{B \times (N+1) \times D},$$

where $t_{\text{cls}}^c$ is a learnable [CLS] token and $D$ is the embedding dimension. The tokens are then passed through a Vision Transformer $f_\theta$ with self-attention layers:

$$\mathbf{T}_c = f_\theta(\mathbf{T}_c^0).$$

#### 3.4.2 CROSS-CHANNEL FEATURE AGGREGATION

We concatenate all channel-specific [CLS] tokens with a global classification token $t_{\text{cls}}^G$:

$$\mathbf{Z}^0 = [t_{\text{cls}}^1, t_{\text{cls}}^2, \ldots, t_{\text{cls}}^C, t_{\text{cls}}^G] \in \mathbb{R}^{B \times (C+1) \times D}.$$

This sequence is processed by a Transformer encoder with $L$ layers:

$$\mathbf{Z} = \text{TransformerEncoder}_L(\mathbf{Z}^0),$$

where multi-head self-attention captures dependencies across channels, and $t_{\text{cls}}^G$ serves as the fused global representation.

#### 3.4.3 MASKED TOKEN RECONSTRUCTION

For masked pretraining, let $M_c \subset \{1, \ldots, N\}$ denote the indices of masked patches in channel $c$. The masked token sequence is defined as:

$$\mathbf{X}_c^{\text{mask}} = \begin{cases} t_i^c & i \notin M_c, \\ t_{\text{mask}} & i \in M_c, \end{cases}$$

where $t_{\text{mask}}$ is a learnable mask token. The masked tokens are decoded using a Transformer decoder conditioned on the fused global representation $t_{\text{cls}}^G$:

$$\hat{\mathbf{X}}_c^{\text{mask}} = \text{TransformerDecoder}(\mathbf{X}_c^{\text{mask}}, t_{\text{cls}}^G).$$

The reconstruction loss for channel $c$ is defined as:

$$\mathcal{L}_c = \frac{1}{|M_c|} \sum_{i \in M_c} \left\| \hat{t}_i^c - t_i^c \right\|_2^2.$$

The total pretraining loss averages across all channels:

$$\mathcal{L}_{\text{pre}} = \frac{1}{C} \sum_{c=1}^C \mathcal{L}_c.$$

### 3.5 $\Delta\Delta G$ PREDICTION

In the antibody affinity maturation prediction task, we employ the multi-channel Vision Transformer and the Transformer Encoder for multi-channel information fusion from the pretrained model. We freeze the Transformer Encoder, while fine-tuning the Vision Transformer and training the predictor on an antibody–antigen affinity dataset to fit the $\Delta\Delta G$ values.

#### 3.5.1 FEATURE EXTRACTION

Given a mutant complex $\mathbf{X}^m$ and its wild-type $\mathbf{X}^w$, we apply the pretrained encoder in prediction mode to extract latent representations. Both $\mathbf{X}^m$ and $\mathbf{X}^w$ are divided into patches and projected into embeddings $\mathbf{T}_0^m, \mathbf{T}_0^w$. These are passed through the pretrained Vision Transformer:

$$\mathbf{T}^m = \text{ViT}(\mathbf{T}_0^m), \quad \mathbf{T}^w = \text{ViT}(\mathbf{T}_0^w).$$

We concatenate all channel [CLS] tokens with a global token for each input:

$$\mathbf{Z}_m^0 = [t_{\text{cls}}^1, \ldots, t_{\text{cls}}^C, t_{\text{cls}}^G] \in \mathbb{R}^{B \times (C+1) \times D},$$

$$\mathbf{Z}_w^0 = [t_{\text{cls}}^1, \ldots, t_{\text{cls}}^C, t_{\text{cls}}^G] \in \mathbb{R}^{B \times (C+1) \times D}.$$

The sequences are then processed by the Transformer encoder:

$$\mathbf{Z}_m = \text{TransformerEncoder}_L(\mathbf{Z}_m^0), \quad \mathbf{Z}_w = \text{TransformerEncoder}_L(\mathbf{Z}_w^0).$$

The outputs $\mathbf{Z}_m, \mathbf{Z}_w \in \mathbb{R}^{B \times (C+1) \times D}$ are flattened into fixed-length feature vectors:

$$\mathbf{H}_m = \text{Flatten}(\mathbf{Z}_m), \quad \mathbf{H}_w = \text{Flatten}(\mathbf{Z}_w).$$

#### 3.5.2 TRAINING

The mutant-wild pair representation is formed by concatenation:

$$\mathbf{H} = [\mathbf{H}_m \parallel \mathbf{H}_w] \in \mathbb{R}^{B \times 2D}.$$

$\mathbf{H}$ is passed into a multi-layer perceptron (MLP) with nonlinear activation, followed by a linear projection to produce the predicted affinity change:

$$\hat{y} = \text{MLP}(\mathbf{H}).$$

The prediction is optimized using mean squared error (MSE) between predicted and experimental $\Delta\Delta G$:

$$\mathcal{L}_{\text{affinity}} = \frac{1}{B} \sum_{i=1}^B (\hat{y}_i - y_i)^2.$$

Table 1: Performance comparison of different methods. Bold denotes the best results.

| Methods | MAE ↓ | RMSE ↓ | PearsonR ↑ | SpearmanR ↑ |
|---------|-------|--------|-----------|-------------|
| FoldX | 1.364 | 2.027 | 0.491 | 0.526 |
| Flex-ddG | 1.236 | 1.849 | 0.497 | 0.484 |
| Bind-ddG | 1.255 | 1.759 | 0.581 | 0.443 |
| RDE-network | 1.189 | 1.665 | 0.508 | 0.592 |
| GearBind | 1.115 | 1.611 | 0.676 | 0.525 |
| ImageAM | 1.042 | 1.500 | 0.704 | 0.590 |
| **ImageAM + P** | **1.017** | **1.460** | **0.718** | **0.604** |

## 4 EXPERIMENTS

### 4.1 EXPERIMENTAL DETAILS

**Datasets.** To enable the model to learn the distribution of feature values characterizing real protein–protein interaction interfaces, we used the unlabeled MaSIF dataset Gainza et al. (2020) during the pretraining stage. This dataset, curated from the Protein Data Bank (PDB) Bank (1971), contains 5,801 real protein–protein interaction structures. For the downstream affinity maturation task, we fine-tuned the pretrained model and evaluated it on the SKEMPI 2.0 dataset Jankauskaitė et al. (2019). SKEMPI 2.0 is the largest antigen–antibody affinity dataset, consisting of 7,085 experimentally measured binding affinity data points across 348 complexes. Following the common preprocessing steps used in previous works Rodrigues et al. (2019); Liu et al. (2021), we removed ambiguous entries, resulting in 5,625 samples for training and evaluation. In addition, we used FoldX Delgado et al. (2019) to sample the mutant structures based on wild-type templates, thereby obtaining structural information for all mutants as model inputs.

**Experimental Settings.** Consistent with prior work on this dataset Cai et al. (2024), we adopted five-fold cross-validation to train and evaluate our model. This approach is widely used in small-sample biological data modeling, as it enhances the stability and reliability of evaluation while reducing bias from random factors. Specifically, the dataset was evenly partitioned into five non-overlapping subsets. In each fold, four subsets (80% of the data) were used for training over 100 epochs, while the remaining subset (20%) served as the test set to assess the model's generalization. This process was repeated five times, ensuring each subset was used once as the test set. We report the mean performance across all folds as the final evaluation. Additional details on the experimental setup are provided in Appendix C.

**Evaluation Metrics.** We employed four representative evaluation metrics: Pearson correlation coefficient (PearsonR), Spearman's rank correlation coefficient (SpearmanR), Mean Absolute Error (MAE), and Root Mean Squared Error (RMSE). Pearson and Spearman assess trend alignment and ranking consistency, respectively, while MAE measures average deviation and RMSE emphasizes large errors. Together, they provide a comprehensive evaluation of prediction accuracy and robustness for $\Delta\Delta G$.

### 4.2 BENCHMARK COMPARISON

Since sequence-based models, including recent protein language models, perform significantly worse than structure-based models on this task, we exclude them from comparison. We benchmarked against five state-of-the-art structure-based methods. Among them, FoldX Delgado et al. (2019) and Flex-ddG Barlow et al. (2018) are energy-based methods, while Bind-ddG Shan et al. (2022), RDE-network Luo et al. (2023), and GearBind Cai et al. (2024) are deep learning-based. Notably, RDE-network and GearBind also incorporate pretraining strategies tailored to mutations.

As shown in Table 1, deep learning approaches consistently outperform traditional energy-based methods. RDE-network and GearBind achieve stronger results than Bind-ddG due to their mutation-oriented pretraining. Our model achieves the best performance among all baselines, demonstrating the effectiveness of modeling multiple interface surface features. By capturing how subtle mutations

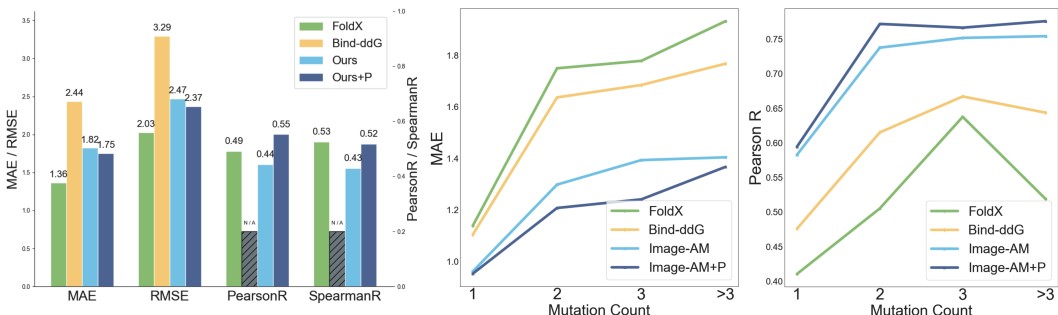

Figure 4: **Left:** Model performance on predicting multi-point mutations after training on single-point mutations, measured by four metrics. **Middle:** Error between model predictions and ground truth for varying numbers of mutations. **Right:** Linear correlation between model predictions and ground truth for varying numbers of mutations.

impact global interface characteristics, our method enables more accurate predictions in antibody affinity maturation.

### 4.3 GENERALIZATION ANALYSIS

**Out-of-Distribution Analysis.** To further evaluate generalization, we designed a more challenging experiment. Each dataset entry involves either single or multi-point mutations. We trained and fine-tuned the model only on single-point mutations, then tested it on multi-point cases. Single-point mutations dominate the dataset (4,059 instances), while multi-point ones are fewer (1,566 instances), making this setup particularly difficult for models trained solely on single-point effects. As shown in Figure 4 Left, Bind-ddG Shan et al. (2022) suffered a large performance drop, even beyond linear correlation evaluation, whereas our method showed much smaller degradation. Furthermore, with pretraining, our model achieved clear improvements, demonstrating the effectiveness of our preraining strategy for generalization.

**Prediction Across Mutation Counts.** Figure 4 Middle and Right illustrates the effect of increasing mutated sites on model performance. On this task, all deep learning methods surpass the empirical FoldX, likely because they learn partial information about multi-point mutations from training data. Pretraining not only improves performance on single mutations but also enhances predictions for more challenging classes. As the number of mutated sites grows, MAE rises markedly, consistent with the larger effects of multi-point mutations, while the Pearson correlation also increases. This is explained by the broader $\Delta\Delta G$ distribution: although absolute errors grow, overall trends become clearer, maintaining or even strengthening linear correlation.

### 4.4 CHANNEL ABLATION STUDY

Our multi-channel input encodes structural and physicochemical features, which collectively yield strong performance. However, the contribution of individual features remains unclear. To investigate this, we evaluated the effect of removing each feature channel.

The results, shown in Figure 5 Left, indicate that all features contribute positively to performance. hydropathy, charge, and RASA have the largest impact, underscoring their importance in this task Srinivasulu et al. (2015); Hebditch & Warwicker (2019). While shape complementarity and hydrogen bonds contribute less, they still play a non-negligible role. For more ablation experiments, please see Appendix A.

### 4.5 SENSITIVITY ANALYSIS OF MASK PATCH SIZE

In this experiment, we studied the impact of patch size, the smallest unit for masking and feature extraction in Vision Transformers, on model performance. Patch size also determines the granularity of information the model can capture from interaction interfaces, which is crucial for performance. As shown in Figure 5 Right, the model performs best with a patch size of 2. This is because smaller

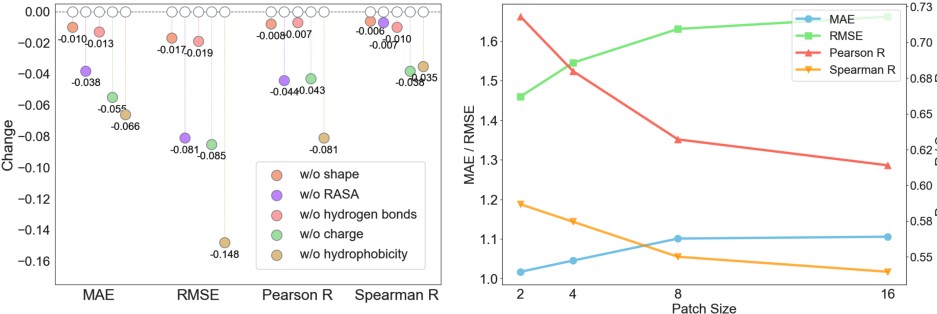

Figure 5: **Left:** Impact of different channels on the overall performance of the model. **Right:** Impact of varying mask patch sizes on model performance.

patches capture finer local features important for protein interfaces, simplify mask reconstruction for more accurate representations, and increase the number of tokens, enhancing the Transformer's ability to model both local and global relationships. Overall, a patch size of 2 provides the best trade-off between granularity, reconstruction difficulty, and modeling capacity.

## 4.6 VISUALIZATION

In this experiment, we visualized the interaction interface of protein 5UFE and its variants after the GA12D, QB25R, EB33G, and DB34G mutations, comparing the model's representations of the wild type and the mutants. Figure 6(A) illustrates the attention weights of the Transformer encoder that integrates multiple features. Unlike the wild type, the model's attention to charge, hydropathy, and shape features shifts significantly in the mutants, indicating that these interface characteristics are notably altered after mutation. Figure 6(B) presents the differences in the charge map between the two types, along with the ViT-generated attention map for this feature, which highlights the regions of the image the model focuses on. As shown, multiple mutations at the interface lead to a clear redistribution of the model's attention. Figure 6(C) further visualizes the interface residues corresponding to the attention map. Interestingly, even for amino acids without apparent structural changes, our model is able to capture shifts in their physicochemical properties. Protein structure visualization was performed using PyMOL DeLano et al. (2002).

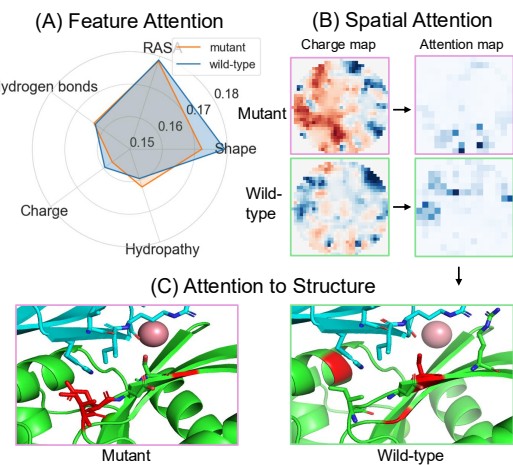

Figure 6: Comparison of protein 5UFE wild type and mutants. **(A)** Encoder attention shows altered focus on interface features, **(B)** charge maps with ViT attention highlight redistributed regions, **(C)** residue-level visualization reveals shifts in physicochemical properties despite minimal structural change.

## 4.7 CONCLUSION

In this work, we introduced ImageAM, a vision transformer–based framework for predicting how amino acid mutations impact antibody–antigen binding affinity. By projecting multiple interface features into 2D images and leveraging a multi-channel masked reconstruction pretraining strategy, ImageAM effectively captures correlations across heterogeneous features and mutation-induced physicochemical changes. Fine-tuning the pretrained encoder for affinity prediction demonstrates that our approach consistently outperforms state-of-the-art methods across various benchmarks and exhibits superior generalization in out-of-distribution scenarios.

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
