

Figure 1: **Left:** Impact of different hidden size on the overall performance of the model. **Middle:** Impact of varying encoder depth on model performance. **Right:** Impact of varying mask ratio in pretraining on model performance.

## A    ABLATION STUDY

To further investigate the impact of the main components on the model's performance, we compared our model with three alternative structural variants: **w/o Interface Feature in Decoder**, where the decoder reconstructs the masked images per channel without utilizing the global interface feature; **w/o Multi-Channel ViT**, where the multi-channel Vision Transformer is replaced by a single ViT shared across all channels; and **w/o Freezing Transformer Encoder**, where the pretrained Transformer encoder is not frozen during downstream prediction of antibody affinity maturation.

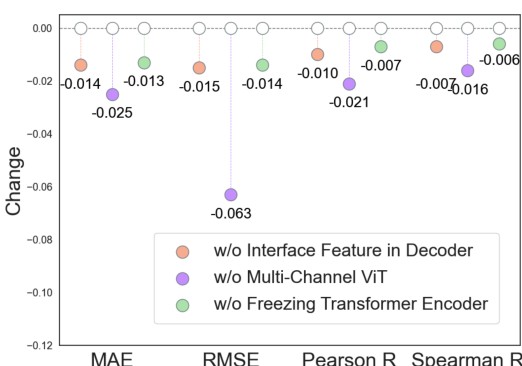

Figure 2: Ablation study on various modules.

From Figure 2, we can draw three key observations: excluding global interface information in the decoder impairs the model's ability to capture holistic interface features, thereby degrading downstream task performance; sharing a single ViT across different feature channels reduces the model's ability to perceive fine-grained distinctions among channels; and freezing the Transformer encoder yields better performance, likely because it preserves the pretrained protein interface features and mitigates overfitting on small datasets.

## B    HYPERPARAMETERS SENSITIVITY

We investigated the effects of mask ratio, hidden layer dimensionality, and the number of Transformer encoder layers on model performance in masked image reconstruction pretraining tasks. These hyperparameters are critical to the model's effectiveness.

As shown in Figure 1, in conventional MAE settings, the typical mask ratio is generally above 50%. However, in our task, the interface images exhibit substantial feature variability, making learning more challenging; thus, the mask ratio cannot be set too high. When the hidden layer dimensionality and the number of encoder layers are small, the model underperforms, likely due to insufficient capacity to capture the diverse information associated with mutations. Conversely, excessively large hidden dimensions or encoder depth lead to performance degradation, which may be attributed to overfitting.

## C    IMPLEMENTATION DETAILS AND HYPERPARAMETERS

The experiments for both the baseline methods and our proposed method are implemented using PyTorch 2.0.1 on a system equipped with 10 Intel(R) Xeon(R) Gold 6248R CPUs and an NVIDIA A100 GPU. Table 1 presents the hyperparameters used for model pretraining ,fine-tuning and evaluation.

Table 1: Hyperparameters

| Hyperparameter | Value |
|---|---|
| Transformer Encoder Layers | 8 |
| Hidden Dimension Size | 32 |
| Learning Rate in Pretraining | 1e-5 |
| Learning Rate in Fine-tuning | 1e-4 |
| Batch Size | 8 |
| Weight Decay in Pretraining | 0.05 |
| Weight Decay in Fine-tuning | 1e-4 |
| Maximum Pretraining Epochs | 20 |
| Number of Transformer Decoder Layers | 4 |
| Mask Patch Size | 2 |
| Mask Ratio | 0.15 |

## D    USE OF LARGE LANGUAGE MODELS

Large Language Models were employed to refine portions of the manuscript, to assist in identifying supporting literature for certain content, and to optimize code used for generating figures and tables.