# OpenReview forum: "Learning Mutation-Aware Visual Context for Antibody-Antigen Affinity Maturation Prediction"
_ICLR.cc/2026/Conference — Submitted to ICLR 2026_

### Official Review · Reviewer_tk1M · 2025-10-29

**Soundness:** 2
**Presentation:** 2
**Contribution:** 1
**Rating:** 2
**Confidence:** 5

**Summary:**

This paper introduces ImageAM, a vision-transformer–based framework that projects antibody–antigen complex interfaces into multi-channel 2D feature maps (encoding geometric, electrostatic, and hydropathy information), and applies a masked image modeling pretraining strategy for mutation-aware affinity prediction. The stated goal is to improve antibody–antigen affinity prediction by learning from unlabeled interface data and capturing mutation-induced structural patterns. However, the work remains conceptually immature and methodologically shallow.

**Strengths:**

1.	Relevant biological motivation:
The problem of predicting antibody–antigen affinity changes is important and timely, especially given the growing interest in antibody design and mutation effect modeling.
2.	Clear high-level intuition:
The paper correctly recognizes that both surface context and mutation locality are critical factors in affinity prediction.
This motivation could, in principle, lead to an interesting cross-domain approach bridging computer vision and structural biology.
3.	Readable overall structure:
Despite weaknesses in clarity and polish, the paper’s organization (motivation → method → experiment → discussion) is relatively easy to follow.

**Weaknesses:**

1. Limitation of Novelty
- The central idea of converting protein surfaces to “images” is not new — it is directly based on MaSIF. Besides, the masked patch pertaining of protein surface has also been proposed in “Surface-vqmae: Vector-quantized masked auto-encoders on molecular surfaces”, and masked pertaining paradigm for DDG prediction has been proposed in PromptDDG.
- Therefore, the only difference is applying a masked image modeling objective (borrowed from MAE). This combination is incremental and not convincingly motivated by biological insight or theoretical reasoning.

2. Poor experimental design
- Only SKEMPI 2.0 is used for evaluation. This is a very small dataset (~7k samples).  Experiments on screening and optimization for certain antigens like SARS-COV-2 is not conducted.
- Misleading correlation metrics. As all the previous methods focus on Per-structure Correlation, this paper seems miss the key metrics. Because the proposed model directly uses protein surface features (e.g., geometric shape, charge, and hydropathy maps), the network can exploit solvent-accessible surface area (SASA) or similar global descriptors that are strongly correlated with absolute binding free energy.
Consequently, the model may appear to achieve a high global correlation simply by ranking antigens according to their average SASA or interface size — without truly learning mutation-level affinity changes within each antigen.
- Lack of strong baselines. Important baselines including DiffAffinity, ProMIM, PromptDDG, LightDDG are not included. The presented metrics seems not as competitive as LightDDG with 0.73+ person R and  0.63+ spearman R.

**Questions:**

See Weakness.

---

### Official Review · Reviewer_gPjN · 2025-10-29

**Soundness:** 2
**Presentation:** 2
**Contribution:** 2
**Rating:** 4
**Confidence:** 4

**Summary:**

This paper presents ImageAM, a vision transformer-based framework for predicting the impact of amino acid mutations on antibody-antigen binding affinity. The key innovation lies in representing the antibody-antigen binding interface as multi-channel 2D projection "images" that encode both structural features and physicochemical properties. The authors employ a masked autoencoder-style pretraining strategy on unlabeled protein-protein interaction data from the MaSIF dataset, where the model learns to reconstruct masked patches across different feature channels. During fine-tuning on the SKEMPI 2.0 dataset, the pretrained encoder is partially frozen and adapted for $\Delta \Delta G$ prediction. The method demonstrates improvements over existing structure-based and energy-based baselines across multiple evaluation metrics, with particularly strong performance on out-of-distribution generalization tasks where models trained on single-point mutations are tested on multi-point mutations.

**Strengths:**

**S1**. The core idea of projecting multiple structural and physicochemical features of protein binding interfaces into 2D image space is both simple and effective, opening an interesting avenue for applying powerful computer vision architectures to protein engineering problems. This representation naturally integrates heterogeneous features that are difficult to combine in graph-based approaches, and the 2D projection enables the use of well-established vision transformer architectures that have proven highly successful in other domains.

**S2**. The paper provides strong empirical evidence for the effectiveness of the proposed approach, demonstrating consistent improvements over multiple baselines. The channel ablation study (Figure 5, Left) provides valuable insights into which physicochemical features are most informative for predicting binding affinity changes.

**Weaknesses:**

**W1**. Vision Transformers, while powerful for general image analysis, do not inherently support SE(3) equivariance, a property that has become increasingly recognized as crucial for structure-based molecular modeling tasks. The 2D projection approach discards this property and the associated symmetries.

**W2**. In the introduction (line 70), the authors characterize ImageAM as *a mutation-aware unsupervised framework*, which is misleading given that the downstream $\Delta \Delta G$ prediction task requires supervised signals from labeled binding affinity data. While the pretraining phase is indeed unsupervised, the overall framework is more accurately described as a self-supervised approach with subsequent supervised fine-tuning.

**W3**. The related work section provides a taxonomic overview of existing methods but lacks critical analysis of their specific limitations and how ImageAM addresses these shortcomings. For structure-based graph methods like GearBind and RDE-network, the authors should provide a more detailed explanation of why graph representations struggle to capture the subtle physicochemical perturbations that their 2D projection approach can detect. Can graph-based models use the proposed features as input node features?

**W4**. While the paper demonstrates that the combination of multi-channel features and ViT encoder yields strong performance, it remains unclear whether the improvements primarily stem from the informative feature representation or from the capacity and inductive biases of the ViT architecture itself. A critical baseline is missing: replacing the ViT encoder with simpler alternatives such as multi-layer perceptrons (MLPs) or standard convolutional neural networks operating on the same multi-channel input. This ablation would help disentangle the contribution of the feature engineering from the contribution of the model architecture, providing clearer insights into which component is more crucial for performance.

**W5**. The benchmark comparison omits two recent and relevant baselines that have demonstrated strong performance on antibody-antigen affinity prediction: AffinityFlow [1] and GeoAB [2]. The absence of these comparisons makes it difficult to assess where ImageAM stands relative to the current state-of-the-art. The authors should either include these baselines in their experimental comparison or provide a clear justification for their exclusion.

---

**Reference**

[1] *C. Chen et al. AffinityFlow: Guided Flows for Antibody Affinity Maturation. ICML 2025.*

[2] *H. Lin et al. GeoAB: Towards Realistic Antibody Design and Reliable Affinity Maturation. ICML 2024.*

**Questions:**

**Q1**.  The paper projects interface features into 2D images, but have the authors considered using 3D volumetric representations instead (i.e., 3D "images" or voxel grids)? While this would increase computational cost, 3D representations could potentially preserve more spatial information and geometric context that is lost during 2D projection. How would the proposed masked reconstruction pretraining strategy extend to 3D convolutional or 3D vision transformer architectures?

**Q2**. The paper uses FoldX to generate mutant structures from wild-type templates. How sensitive are the model's predictions to the quality and diversity of input structures?

**Q3**. The caption and text for Figure 6(B) state that it demonstrates how the attention weights shift between wild-type and mutant charge maps, but the interpretation provided is limited. The authors note that *multiple mutations at the interface lead to a clear redistribution of the model's attention*, but this observation seems somewhat tautological. One would expect the attention patterns to differ between structures with different charge distributions. What specific insights should readers extract from this visualization beyond the obvious fact that mutations change feature maps?

**Details Of Ethics Concerns:**

No concerns

---

### Official Review · Reviewer_Q4F3 · 2025-10-31

**Soundness:** 4
**Presentation:** 4
**Contribution:** 3
**Rating:** 4
**Confidence:** 4

**Summary:**

The paper presents a antibody-antigen interaction prediction method based on interface representation. The authors first pre-train an encoder-decoder framework from multiple surface feature channels and a vision transformer (ViT) featurizer. The learned representation is then used to predict the impact of mutations by using representations of wild-type and mutant structures. The experimental results show improved performance against the baseline methods, with good generalization and interpretability.

**Strengths:**

- The formulation of the method is overall solid and clearly presented.

- The model shows clear performance improvements compared to the baselines.

- Interpretation is straightforward and meaningful. The learned spacial attention patterns highlight biologically relevant structural features.

**Weaknesses:**

- The proposed method is technically solid but limited application is presented. Both the encoder-decoder architecture and learned embedding could have wider implications in protein interface representation and design. The paper only focuses on one task (ddG prediction) which somewhat narrow. See Questions.

**Questions:**

- Could the model be used to screen or select potential binders (eg antibodies) with enhanced affinity?

- Since the model relies on pre-computed features, how efficient is the computation and how transferrable is it to the guidance of generative models for protein designs?

- How does using MaSIF features compare to simpler featurization such as voxels, atom occupancy by element types, or atomic adjacency graphs?

- Since the model relies on pre-computed features, how efficient is the computation and how transferrable is it to the guidance of protein designs?

Minor:
- Line 345: SKEMPI 2.0 is not limited to antibody-antigen but general protein-protein interaction

- The citations are not properly rendered (no brackets)

---

### Meta-Review · Area_Chair_yZEA · 2026-01-06

**Summary:**

The paper proposes ImageAM, a vision transformer model that encodes multiple structural and physicochemical features of antibody–antigen interfaces as 2D images, and uses multi-image masked reconstruction pretraining followed by fine-tuning to predict mutation effects on binding affinity. The authors report improvements over existing methods on standard affinity change benchmarks.

**Reviewer Concerns:**

Reviewers broadly agreed that the work’s novelty is limited. The core idea is an adaptation of existing vision transformer techniques rather than a fundamentally new machine learning method. Experimental evaluation was criticized for insufficiently strong baselines, lack of ablation studies, and limited statistical analysis.

**Reviewer Scores:**

The authors did not submit a rebuttal, and I do not expect any reviewer scores to change.

---

### Decision · Program_Chairs · 2026-01-26

Reject